# OpenReview forum: "The Biased Artist: Exploiting Cultural Biases via Homoglyphs in Text-Guided Image Generation Models"
_ICLR.cc/2023/Conference — Submitted to ICLR 2023_

### Official Review · Reviewer_GVzw · 2022-10-23

**Confidence:** 3
**Correctness:** 3
**Technical Novelty And Significance:** 3
**Empirical Novelty And Significance:** 3
**Recommendation:** 5

**Clarity, Quality, Novelty And Reproducibility:**

Clarity: The paper is well-written and easy to follow.

Quality: The analysis of the phenomenon of bias is thorough enough. But the analysis of the reason is weak and potential solutions are too naive.

Novelty: The problem itself is interesting. But samely, the proposed solutions are trivial and intuitive.

Reproducibility: Good.

**Strength And Weaknesses:**

Strengths:
1) The paper studies the text-to-image models from an interesting perspective.
2) The findings presented in the paper are interesting.
3) The authors did a thorough analysis to draw these conclusions.

Weakness:
1) My major concern is that the proposed solutions are too simple and intuitive. In a paper, we hope to see how to alleviate the problem using a more technically interesting solution rather than merely present the problem.
2) Reasons for the model behaviour are just verbally explained and they are not verified by the experiments.

**Summary Of The Paper:**

The paper investigates the impact of homoglyph replacements of the texts on text-to-image models. Three interesting observations are found:

1) Homoglyph replacements can cause image content obfuscation.

2) Homoglyph replacements can induce cultural bias.

3) Homoglyph susceptibility can be observed in other text-to-image models, e.g., Stable Diffusion. However, the effect is weaker in Stable Diffusion than that in DALLE-2.

The authors also suggest two simple solutions to avoid biases. One is to replace all characters in the texts with standard characters, the other is to train the language encoders with multilingual texts.

**Summary Of The Review:**

I tend to give a score of 5 to this paper.

I think the topic this paper explores is interesting and the analysis of the phenomenon is thorough. My concerns are: 1) the paper didn't dive into the reason behind the phenomenon; 2) The proposed solution is too simple.

---

> ### Author Response · Authors · 2022-11-14
> **Response**
>
> We thank the reviewer for the time spent reviewing our paper. We are also thankful for the provided feedback and the raised discussion points.
>
> 1.) Proposed solutions are too simple: We added a novel homoglyph unlearning approach in Sec. 4.3 as a more elaborated technical solution that directly increases the robustness of pre-trained text encoders to homoglyphs. So without training a separate text encoder from scratch or fine-tuning the generative model - both procedures are data, time and computation intensive - our approach allows to remove most of the biasing behavior from an encoder in a reasonable amount of time (about 45 Minutes for the stated experiment). Still, we believe that a solution does not always need to be technically complicated but simple yet effective solutions should rather be preferred to allow easy implementation. We believe by pointing out this simple approach, we offer service and model providers a fast and easy-to-implement solution to avoid such homoglyph manipulations.
>
> 2.) Reasons for model behavior: We cite here our answer to Reviewer 4KN7: Analyzing the influence of the text encoder and the generative model on their own and in combination probably requires re-training parts or even the full models with multilingual datasets. Unfortunately, this is computational, economical, and energy-related not feasible. Furthermore, Stable Diffusion is not compatible with the M-CLIP model since it has been trained with a specific OpenAI CLIP model. So simply replacing the text encoder with another model is also not possible. However, we believe that the text encoders already are fully capable of "biasing" the image generation process and act as the biasing signal in the process. As we have shown for CLIP-based text encoders (Sec. F.1) on their own show already sensitive behavior in the presence of non-Latin characters. Since most text-to-image synthesis models are based on their text embeddings for guidance, biasing the embedding space is already sufficient to manipulate image generation. With our homoglyph unlearning approach (Sec. 4.3), we demonstrated that removing the biasing signal from the encoder is sufficient to avoid biases. However, we believe that diffusion models also learn to understand fine-grained text embedding details and probably learn similar encoding sensitivity.

---

> > ### Author Response · Authors · 2022-11-28
> > **Looking forward to receiving your feedback**
> >
> > Dear Reviewer GVzw,
> >
> > as you may know, we are approaching the deadline for the final recommendation. We would therefore be very grateful if you could provide feedback on our rebuttal. We also encourage you to see the additional quantitative results and particularly our homoglyph unlearning procedure as a technical solution to increase a model's robustness against homoglyphs. If you are satisfied with our response and changes, please consider updating your rating. If you have any additional questions, please feel free to contact us.
> >
> > Best regards,
> > Authors

---

### Official Review · Reviewer_k5Mu · 2022-10-24

**Confidence:** 4
**Correctness:** 4
**Technical Novelty And Significance:** 2
**Empirical Novelty And Significance:** 2
**Recommendation:** 3

**Clarity, Quality, Novelty And Reproducibility:**

The paper is well written and easy to understand.
The findings are somewhat novel and easy to reproduce.

**Strength And Weaknesses:**

The paper is very clear and shows several examples for how exchanging characters with similar other characters affects the generate image, either by completely ignoring important parts of the caption (e.g. noun phrases if a single character was changed) or by biasing the output towards different contents based on the origin of the inserted character (e.g. Indian or Vietnamese portrait).

This is a somewhat interesting finding since it indicates that the models were trained at least with some of these non-Latin characters and learned to associate those with certain contents (e.g. geographic regions).

While the finding itself are interesting I don't think they are necessarily surprising. While the characters may look similarly the underlying unicode encoding for those characters is discrete and, therefore, the language model would treat those as independent characters, no matter what they look like visually.

I don't think this can be called an adversial attack though. IMO an adversarial attack is an attack that is a) indistinguishable from a normal "correct" input for a human and b) not easy to detect automatically. While a) may be true in this case, the authors themselves state that it is easy to detect these changes simply by checking the unicode encodings of the input string.

Also, I'm also not sure if calling this behavior "biased" is correct. If I ask for a portrait of an "Indian woman" getting a portrait of an Indian woman is not biased behavior, but desired behavior. Similarly, replacing a Latin character in the caption with an Indian character can be interpreted as generating an image with Indian influence.

Overall, I feel like this problem (if it really is a problem) is easily addressed by standard preprocessing, similarly to converting all text to lower-case, non-Latin unicode characters can be mapped to their closest unicode characters or, alternatively, the model can be trained on more non-Latin unicodes.

**Summary Of The Paper:**

The shows that current text-to-image models such as DALLE2 and Stable Diffusion produce different images even if only a single character of the text description is changed to a (visually similar) different character (e.g. replacing the latin character "A" with the Greek character capital alpha "Α"). This behavior also shows certain biases of the underlying models, e.g., generating images coming from the cultural or geographic background based on the replace character (e.g., whether a Latin character is replaced by a Greek or a Arabic character). This behavior allows users to change the model output while seemingly keeping the text input constant (the unicode of the character changes but visually this is not obvious when reading the caption).
The authors argue that this behavior may allow attackers to reduce the generation quality of a model or to portrait the model behavior differently by imperceptibly changing the input caption.

**Summary Of The Review:**

While the findings are interesting I don't think they contain enough novelty or impact to accept the paper in its current form.
Also, I am not sure about the terminology used in the paper, e.g., calling these changes "adversarial attacks" or claiming that this behavior is evidence of model bias.

---

> ### Author Response · Authors · 2022-11-14
> **Response**
>
> We thank the reviewer for the time spent reviewing our paper. We are also thankful for the detailed review and the raised discussion points.
>
> 1.) Results are not necessarily surprising: Since computer systems process characters based on their underlying encodings and not based on their visual appearance, it might not be surprising that the systems interpret homoglyphs in different ways. However, the surprising insight of the paper is that the models learn to connect characters from specific encoding scripts with their local culture and then reflect this connection in their generated images. And even subtle changes, such as replacing a single 'o' in 'of' with non-Latin characters, are already sufficient to influence the image generation process in a non-random but culturally influenced way.
>
> 2.) It is not an adversarial attack: Please note that we do not directly phrase our findings as an adversarial attack in the sense of crafting adversarial examples traditionally, even if some concepts are similar. We rather mainly emphasize that current models are sensitive to single character encodings, which might be misused in some adversarial settings to obfuscate content (particularly as stated in (now) Appx. F.1) but also stands on itself as an inherent behavior of multi-modal models.
>
> 3.) Calling it a "bias": We chose the term "bias" since querying the models with a standard, Latin-only prompt (e.g. "photos of women") generates images depicting a kind of average representation of the prompt (partly ignoring the potential Western bias present in the training data). If then changing a single character without any explicit statement of cultures leads to the generation of images representing no longer the average looks but rather some stereotypes, e.g., Korean or Indian appearance, we think calling this behavior a "bias" is appropriate since it directs the image generation process in a specific direction.
>
> 4.) Manipulations could be simply detected: We agree that a homoglyph detector probably prevents homoglyph manipulations in an adversarial setting, as we also discussed in Section 5.2. However, since models like Dall-E 2 and Stable Diffusion have no such defense implemented or even discussed this possibility, it shows that current systems are not aware of or ignore this kind of attack.
> Therefore, we think informing the community about this behavior is important to motivate the implementation of exactly such defense mechanisms as a first step or at least notify the users of the possibility.

---

> > ### Author Response · Authors · 2022-11-28
> > **Looking forward to receiving your feedback**
> >
> > Dear Reviewer k5Mu,
> >
> > as you may know, we are approaching the deadline for the final recommendation. We would therefore be very grateful if you could provide feedback on our rebuttal. We also encourage you to see the additional quantitative results and homoglyph unlearning procedure that have been added to further strengthen our paper. If you are satisfied with our response and changes, please consider updating your rating. If you have any additional questions, please feel free to contact us.
> >
> > Best regards,
> > Authors

---

### Official Review · Reviewer_wkRc · 2022-10-25

**Confidence:** 5
**Clarity, Quality, Novelty And Reproducibility:** There's no quantitative result in the…
**Correctness:** 1
**Technical Novelty And Significance:** 2
**Empirical Novelty And Significance:** 2
**Recommendation:** 1

**Strength And Weaknesses:**

The strength of this paper is that the situation assumed in the paper is very unique. The paper also shows that by using a character in a language, one can obtain an image that belongs to that cultural group. It may be trivial but I don't think anybody has shown it.

Several weaknesses
1. This is a 100% qualitative paper with no attempt to quantify the extent of the problem being discussed. The problem of bias is essentially finding the right (fair, for example) measure for the problem. We can't just cherry pick a few examples and claim that we found something meaningful. This needs to be thoroughly measured and validated.

2. The attack scenario, while novel, doesn't sound reasonable to me. When will this happen? Let's say there's an attacker, a malicious app. A user enters a text input. The app will then modify the query and inject some signals there. But then why would the app show the modified text back to the user using a non standard latin character? The whole point about homoglyphs is that humans can be confused, but not machines. Why wouldn't the app just modify the text more explicitly by adding more words and not show the modified text to users?

**Summary Of The Paper:**

This paper addresses an attack scenario for text guided image generation models like dalle2 or text-image joint model like clip. This paper is entirely qualitative and doesn't provide any quantitative metrics. The paper argues that an attack can be made to these models by replacing one alphabet character in an input text with a unicode non-latin character such as a Korean character, which will affect the output image.

**Summary Of The Review:**

This paper proposes to study an interesting behavior of image-text generation/joint models. Due to the lack of scientific measures to validate the argument, this paper is not ready for publication at a venue like iclr.

---

> ### Author Response · Authors · 2022-11-14
> **Response**
>
> We thank the reviewer for the time spent reviewing our paper and for the discussion points raised.
>
> 1.) 100% Qualitative Paper: We kindly emphasize that the ICLR submission guidelines do not exclude qualitative results/papers from submission. Moreover, the Reviewer Guide (https://iclr.cc/Conferences/2023/ReviewerGuide, "Reviewing a submission: step-by-step") states "3.4: Submissions bring value to the ICLR community when they convincingly demonstrate new, relevant, impactful knowledge (incl., empirical, theoretical, for practitioners, etc)." We believe that our findings indeed provide insightful empirical knowledge, even without an extensive quantitative evaluation. Even if the behavior would only be present in a small number of cases, we think that this model behavior is still a meaningful finding because it illustrates that text-guided generative models are highly sensitive to single-character encodings. And to show this behavior, no extensive quantitative evaluation is necessary. Also, we want to highlight that we did not cherry-pick our results but tested our findings on various concepts, models and scripts!
>
> However, to provide a quantitative analysis, we introduce relative bias as a metric to quantify the biasing effects of various homoglyphs from different scripts. We evaluated the biasing effects of homoglyphs from five different scripts on three different concept groups. Please see Sec. 3 for the introduction of our metric and Sec. 4.2 and 4.3 for the evaluation results.
>
> 2.) Attack scenario: Let us sketch the following scenario: A user uses some automatic prompt tools, e.g., Dallelist and Write AI Art Prompts, to improve her/his prompts. These tools can add additional keywords or change existing words, similar to a spell-checking tool. An adversary might then provide a malicious tool that injects homoglyphs into a user's prompts, or even directly a malicious prompt through prompt collections like lexica.art. Take then, for example, the OpenAI Dall-E 2 API. The adversary has no direct access to the API or the website but only the user's prompt through the browser plugin. Therefore, the user can see the full prompt before starting the generation process. If the malicious tool simply injects explicit trigger words or descriptions, then the user probably notices the manipulation. If, however, only a single character is replaced by a homoglyph, the user can only hardly detect the manipulation.
>
> 3.) Clarity, Quality, Novelty and Reproducibility: "There's no quantitative result in the paper.". Only because a paper does not provide extensive quantitative results (which we added now), we think the clarity, quality, novelty and reproducibility of our paper could still be assessed. All other reviewers agree that our paper is well-written and studies an interesting behavior of image-to-text synthesis models.
>
> 4.) Correctness: "The main claims of the paper are incorrect or not at all supported by theory or empirical results.". Please provide us with more detailed information about which claims of the paper are not supported by "theory or empirical results". We believe this rating does not honestly reflect the paper's content and contributions.
>
> General remark: Please note that other reviewers have acknowledged the novel contributions and intriguing insights of our paper. We hope that our introduced changes also eliminate the stated weaknesses. We thus would urge the reviewer to reconsider the given score. For further questions or uncertainties, please do not hesitate to contact us.

---

> > ### Comment · Reviewer_wkRc · 2022-11-18
> > **Evaluation**
> >
> > Thank you for providing a response. I don't agree that this paper doesn't need a quantitative evaluation. There are papers in which their hypotheses can be proven without quantitative analysis (e.g. via theoretical proof). This is not the case with the current paper which reports empirical findings. In particular, the topic of ML bias heavily relies on the careful definition of bias and systematic and objective measurements. The claim that "no extensive quantitative evaluation is necessary" sounds surprising to me. What is the testable hypothesis of this paper? It says image-text models "learn cultural biases." How can we test this hypothesis without quantitative statistical analysis? Hence, my previous rating "The main claims of the paper are incorrect or not at all supported by theory or empirical results."
> >
> > Having said that, I appreciate the authors for adding a quantitative measure. I still don't feel this is enough. Fig 5 seems to show that the "relative bias" is reduced after treatment, but the same can be achieved by random models (0 correlation). These bias-metrics should be presented with the main measures (image generation in this case) such that it can show the proposed method can reduce bias while retraining the original model performance in the main task. This is very typically done in the literature on ML bias mitigation.
> >
> > About the attack scenario -- I'm not sure if the situation described here is very realistic or relevant to model learning. In addition, "If, however, only a single character is replaced by a homoglyph, the user can only hardly detect the manipulation." --> Did you validate this assumption that the user can't recognize the single character difference?
> >
> > The notion of cultural bias in this paper is the similarity between image content and character encoding. This similarity is very well expected and may be even beneficial to users. What are the actual harms? The relative bias measure used in Fig 5 is just similarity. Why is this expected similarity called out and needs mitigation which may limit the model performance? I think more careful justifications are necessary.

---

> > > ### Author Response · Authors · 2022-11-23
> > > **Response 2**
> > >
> > > We appreciate the reviewer's more detailed feedback and clarification. Please excuse that we kindly and slightly disagree on some of the points.
> > >
> > > "Fig. 5 seems to show that the "relative bias" is reduced after treatment, but the same can be achieved by random models (0 correlation).". Maybe we misunderstand the point here, but a random model with randomly initialized weights would be useless for any task. Our approach to reduce the cultural bias by fine-tuning the text encoder to become invariant to homoglyphs maintains its overall utility, and the whole model is still able to generate photorealistic images following any input prompts. Please note that the Fréchet inception distance (FID) score was already provided in the paper. The FID is a metric used to assess the quality of images created by a generative model, like diffusion models or GANs. Unlike the earlier inception score, which evaluates only the distribution of generated images, the FID compares the distribution of generated images with the distribution of a set of real test images (MS-COCO dataset) [3]. Since the FID score stays roughly the same as the standard model in our experiments, we exactly show what you ask for with the additional results (see Sec. 4.3). We also want to emphasize that we measured this bias in the context of the main task, i.e., image generation, and showed that homoglyphs lead to the generation of images that depict content related to the homoglyph's cultural background, measured by a large CLIP model.
> > >
> > > To summarize, since we added quantitative metrics and empirical results, we believe "main claims of the paper are incorrect or not at all supported by theory or empirical results" is a no longer fair rating for our work and should be reconsidered.
> > >
> > > We also want to point out that "retraining the original model" is computationally not feasible. Stable Diffusion has been trained on 256 A100 GPUs [1] for 150,000 hours. Retraining such a model would not only produce unbearable costs but also consume a huge amount of power and emit tons of carbon. That is why our fine-tuning approach is preferable since it does not need large training times but can be done in minutes while maintaining the model's overall utility.
> > >
> > > "Did you validate this assumption that the user can't recognize the single character difference?"
> > > Since the rendering of distinct characters is highly dependent on the font used in an application, there probably exists no objective measurement for specific homoglyphs. ꓧowever, it is obvious thаt sοme homoglyphs look nearly identical and are hard tо spot, еvеn without a user study perfоrmed. This point is also demonstrated by the last sentence in which we inserted various homoglyphs. Indeed, it is an interesting question how humans perceive homoglyphs. However, homoglyphs are already a well-known phenomenon and are already used, e.g., in adversarial examples, backdoor attacks, domain-name spoofing or phishing attacks in various contexts, see for example [4 - 8]. Our work extends the application of homoglyphs to the context of text-guided generative models.
> > >
> > > "Why is this expected similarity called out and needs mitigation which may limit the model performance"
> > > If we take a look at recent news on applied AI, e.g., automatic video or image labeling, there have been many reports on biases against people of color or general gender biases [2]. Now, if a single homoglyph can be used to force the creation of images that reflect certain cultural backgrounds, without the user knowing about the manipulations, one could imagine many ways to discriminate against people or damage the model's reputation. For example, an adversary could force that images depicting "working farmers" will always have people with dark skin color or "ugly people" will always have Asian looks, which definitely reflects racist stereotypes. So there are many reasons why such a model behavior might not be beneficial. We believe that the reasons we have presented, clearly demonstrate why homoglyphs and the general cultural bias of the models pose a huge risk and why the whole topic including possible mitigation strategies should be discussed. However, we also discussed the point of whether it is a vulnerability or can also be a feature in Sec. 5.3.
> > >
> > > "This similarity is very well expected". We still do not agree with this point, and also emphasize that the other reviewers agree that it is a surprising and interesting model behavior. Particularly since it was claimed that the models (DALL-E 2 and Stable Diffusion) have been trained on English-only data and not multilingual data.
> > >
> > > We hope we could clarify the importance of the problem of cultural biases, which are highly influenced by single homoglyphs in prompts, presented in our paper and why it is important to inform the community about this unexpected model behavior.

---

> > > > ### Author Response · Authors · 2022-11-23
> > > > **Response 3**
> > > >
> > > > [1] https://huggingface.co/CompVis/stable-diffusion-v1-4
> > > > [2] https://www.nytimes.com/2021/09/03/technology/facebook-ai-race-primates.html
> > > > [3] https://en.wikipedia.org/wiki/Fr%C3%A9chet_inception_distance
> > > > [4] Wolff. Attacking Neural Text Detectors, ICLR 2020 Workshop Towards Trustworthy ML: Rethinking Security and Privacy for ML
> > > > [5] Boucher et al. Bad characters: Imperceptible NLP attacks. IEEE S&P 2022
> > > > [6] Li et al. Hidden backdoors in human-centric language models. CCS 2021
> > > > [7] https://cisomag.com/homoglyph-attacks/
> > > > [8] https://businessinsights.bitdefender.com/homograph-phishing-attacks-when-user-awareness-is-not-enough

---

### Official Review · Reviewer_4KN7 · 2022-10-25

**Confidence:** 4
**Correctness:** 3
**Technical Novelty And Significance:** 3
**Empirical Novelty And Significance:** 4
**Recommendation:** 6

**Clarity, Quality, Novelty And Reproducibility:**

I think the paper is overall well-written. The ideas are presented clearly. To the best of my knowledge, the proposed attack is novel and the effectiveness of it is surprising. The authors provide the source code to reproduce the results. I strongly encourage open-sourcing the code upon acceptance.

There is a point that I don't understand well. Figure 5 seems to be making an even stronger point that the rest of the paper: the sensitivity appears to be in the character level, even independently of *where* the character is in the sentence. My understanding of the way the authors form the embeddings in Figure 5 is the following: i) pass the entire sentence through the text encoder, ii) pass the character to be replaced from the text-encoder, iii) pass the replacement from the text-encoder and finally iv) form the embedding by taking the sum of i) + iii) - ii). Is that the case? If that's true, there is no position information on where the character to be replaced is, e.g. think of what happens when it appears two times.

Minor typo: Athene -> Athens in Section 4.2.



**Strength And Weaknesses:**

Strengths:

*  The paper is definitely interesting. The fact that a single letter change can have such a dramatic impact on the generation quality was surprising to me.
* The authors show that this phenomenon is not specific to one model. There seems to be a more universal issue.
* The observation that M-CLIP is more robust is very insightful. This suggests that multilingual training could reduce cultural biases present in text-to-image generative models.



Weaknesses:

* I am not sure about whether this observation poses a new security threat for these models. If a single letter change introduces such cultural biases, I am fairly certain that more obvious changes to the prompt would also trigger culturally biased generations. If the goal of the attacker is to use Stable Diffusion or DALLE-2 to generate disturbing images, I think there are other ways to achieve this as well. So the question becomes, why the homoglyph replacement attack is more dangerous than other types of attacks. I am not sure I have a good answer on that. Since the attacker is the one deciding the prompt in the first place, why is it important to make it look innocent to someone else? How could it be maliciously used? I would love to hear more from the authors on this.
* I strongly disagree with the potential interpretation of this phenomenon as a feature in Section 5.3. The users of an API should not have their prompt manipulated before entering the system, unless it is explicitly documented or/and there is an option to deactivate it. It is one thing to use filters to catch harmful prompts and another thing to fool the users by presenting results for a modified prompt. But even if the intent is good and the users are informed about how their input was modified, it is still very unclear how the reduction of societal biases would be achieved. What is the right amount of homoglyphs to replace for a "fair" generation? How does one ensure that the new generations are not having new biases? The authors rightfully acknowledge that the method works better for specific cultures.
* The authors propose multilingual training as a potential solution to the observed problem. However, DALLE-2 seems to have been trained on multilingual data. I tried a couple of prompts in Japanese and German and the system successfully generated pictures corresponding to the prompts. So is the claim that the model should have been trained on *more* multilingual data? Or was the text-encoder frozen during the training and we need to train the text-encoder on multilingual data?
* I am fairly confused about whether the issue is in the Text Encoder, the generative model or both. See also the bullet point above. Would love some experimentation to understand this better.

**Summary Of The Paper:**

The paper shows that the quality of the generated images from text-to-image models such as DALLE-2 and Stable Diffusion can be greatly impacted by single homoglyph replacements. A malicious user can replace a character with a homoglyph that looks the same to the human eye and the text-to-image system produces images with cultural biases. The authors show that this behavior is (to a certain extent) consistent between different text-to-image models. Finally, the authors propose multilingual training of the text-encoder as a potential solution to this problem.

**Summary Of The Review:**

This paper presents a surprising observation: single homoglyph replacements change dramatically the quality of the samples and lead to biased generations. The fact that generative models amplify societal biases is not new. This paper shows a new aspect of this problem: the sensitivity to small input perturbations. I believe this is an important observation - even if a system appears to have diverse outputs, a small change to the input can drastically deteriorate its performance.

I am not convinced that the attack itself poses a new security problem. However, it broadens our understanding of the current limitations of these models. Therefore, I (weakly) recommend acceptance for this paper.

I encourage the authors to engage in the rebuttal and address the questions raised above.

---

> ### Author Response · Authors · 2022-11-14
> **Response 1**
>
> We thank the reviewer for the time spent reviewing our paper and for the detailed review. We are also thankful for the questions and discussion points raised.
>
> 1.) Security threat model: While we focus on hard-to-spot homoglyph replacements, we do not exclude the possibility of stronger attacks by inserting more obvious changes to the text prompt. However, our setting assumes that the prompt that is visible to the user is the same prompt that is processed by the models and that no additional adversarial intermediate step, e.g., additional prompt manipulation, is performed. Recently, automatic prompt tools, e.g., Dallelist and Write AI Art Prompts, gained popularity among non-professional users and offer to enhance user prompts by automatically providing word substitutions or additional phrases. Since these tools are also available as browser plugins and can directly change the prompt text, a malicious tool could easily insert non-Latin characters without the user noticing any difference. Other manipulations, which are not based on homoglyphs, would be easy to spot by a user, so it is important that adversarially speaking, the manipulations are invisible. We also emphasize that Dall-E 2 and Stable Diffusion are not only used by researchers but more and more by non-professionals, who do not expect some adversarial manipulations. By then biasing the generation or even rendering it useless might give the user a false sense of the models quality, which might not only lead to bad publicity (including criticism of the bad quality and racial biases) but also economic damage, particularly for chargeable services, such as DALL-E 2.
>
> 2.) API prompt manipulations: Please note that we do not propose that APIs should manipulate users' prompts without their consent or awareness. In contrast, we meant by "an API might also offer to automatically include such cultural biases in the prompt." (Section 5.3) that the APIs could offer users an option to explicitly allow the API to guide the generation with some homoglyph injections. For example, a user from an Asian country might explicitly allow injecting a cultural bias associated with her/his home country. We agree on the point that the systems should never do such things without a user's consent! As a consequence, we rewrote the second part of the sections to make our point clearer.
>
> Regarding the question of the right amount of homoglyphs inserted for a fair generation: We think there might not be a "right" amount of homoglyphs but still requires some testing for each use case. However, also for standard prompts the user has to try different combinations of words and phrases to generate images depicting the imagined content. We also think such homoglyph replacements will not solve the overall problems of (Western) biases in multimodal systems but that they can rather act as an interesting feature in such systems and allow, to some extent, to visualize existing biases.

---

> > ### Author Response · Authors · 2022-11-14
> > **Response 2**
> >
> > 3.) Multilingual training as a solution: As the DALL-E 2 paper [Ramesh et al. Hierarchical Text-Conditional Image Generation with CLIP Latents, 2022] states, DALL-E 2 freezes the CLIP model during training of the prior and decoder. Without exact training details stated, we assume that the CLIP text encoder is a pre-trained model, such as available at https://github.com/openai/CLIP, since only the training of the image decoder and prior models are specified. CLIP itself has been trained on a non-public dataset of 400M image-text pairs with titles and descriptions in English (see [Radford et al., Learning Transferable Visual Models From Natural Language Supervision, 2021], Section 2.2). Since the dataset is not available, we cannot make any statements about how much multilingual data is available. However, as our results on multilingual CLIP demonstrate, such models are more robust against different character encodings than the standard CLIP model. So yes, we believe training on a larger set of non-Latin characters indeed increases a model's robustness.
> >
> > 4.) Experiments on the influence of the Encoder / Diffusion model: Analyzing the influence of the text encoder and the generative model on their own and in combination probably requires re-training parts or even the full models with multilingual datasets. Unfortunately, this is computational, economical, and energy-related not feasible. Furthermore, Stable Diffusion is not compatible with the M-CLIP model since it has been trained with a specific OpenAI CLIP model. So simply replacing the text encoder with another model is also not possible. However, we believe that the text encoders already are fully capable of "biasing" the image generation process and act as the biasing signal in the process. As we have shown for CLIP-based text encoders (Sec. F.1) on their own show already sensitive behavior in the presence of non-Latin characters. Since most text-to-image synthesis models are based on their text embeddings for guidance, biasing the embedding space is already sufficient to manipulate image generation. With our homoglyph unlearning approach (Sec. 4.3), we demonstrated that removing the biasing signal from the encoder is sufficient to avoid biases. However, we believe that diffusion models also learn to understand fine-grained text embedding details and probably learn similar encoding sensitivity.

---

> > > ### Comment · Reviewer_4KN7 · 2022-11-18
> > > **Rebuttal Acknowledgment**
> > >
> > > I want to thank the authors for their clarifications. For the time being, I maintain my recommendation which is currently "6: marginally above the acceptance threshold".

---

### Author Response · Authors · 2022-11-14
**Thank you for reviewing**

We sincerely thank all the reviewers for their time and effort spent in evaluating our submission. We are particularly grateful for pushing our work towards quantitative evaluations and hope that our introduced changes eliminate existing concerns.

Here, we address the major changes and further respond to individual questions and remarks under the respective reviews with more details. We marked the updates and changes in the paper with blue color.

1.) To quantify the influence of homoglyphs, we introduced the Relative Bias (Sec. 3, "Quantifying The Influence of Homoglyphs") as a novel metric. We further created three datasets for different cultural concepts to differentiate between the corresponding biasing effects on these concepts. The results for Stable Diffusion are stated in Fig. 5. The findings are discussed in Sec. 4.2 and 4.3.

2.) As a technical solution for the susceptibility to homoglyph manipulations, we proposed a novel and effective homoglyph unlearning procedure in Sec. 4.3. We will make the code publicly available if the paper gets accepted.

3.) To keep the page limit, we moved the section "Image Content Obfuscation" to Appx. C and the section "Non-Generative Multimodal Models Behave Similarly Sensitively" to Appx. F.1.

Please feel free to reach out to us if you need any further clarification or have any additional questions.

---

> ### Author Response · Authors · 2022-11-18
> **Thank you for reviewing**
>
> Dear reviewers and AC,
>
> we have put a lot of effort into responding to the questions and comments raised in the individual reviews and improving our paper. We hope that our responses satisfactorily answer all the points raised and lead to an update of the scores. Unfortunately, we have not yet received any response from the reviewers. As we are heading close to the revised manuscript submission deadline, we would appreciate the reviewers' feedback on our rebuttal.
>
> Best
> the authors

---

### Decision · Program_Chairs · 2023-01-20

**Decision:**

Reject

**Justification For Why Not Higher Score:**

Despite discussion and rebuttal from the authors, reviewers were unconvinced by the attack scenario addressed in the paper.

**Justification For Why Not Lower Score:**

N/A

**Metareview: Summary, Strengths And Weaknesses:**

This paper presents the observation that text-to-image generation models are sensitive to specific Unicode character sets within text description when generating images. Because different alphabets share visually similar characters (homoglyphs), one could for instance use a non-Latin character (e.g., from the Korean or Indian alphabet) and steer the image generator towards a “cultural” subset of the latent space while misleading a user who has copy-pasted that prompt while believing it is in the Latin alphabet. The authors conduct a series of explorations with several text-to-image generators (such as DALL-E 2 and Stable Diffusion 1.4) for a variety of prompts and alphabet injections, for which they comment themselves on the biases, and report numerical similarity scores on CLIP embeddings of a target image and text prompts with alphabet injection. Finally, they inspect the LAION dataset for presence of various alphabet letters. The authors also comment on the inherent bias in data used to train the image generator models, and specifically the notion of “white” or “Northern American” as a norm.

Reviewers praised the interesting problem and findings (GVzw, k5Mu, wkRc, 4KN7). Some reviewers praised thorough analysis (GVzw, 4KN7) and the observation that multilingual training in M-CLIP reduces the observed effect.

Weaknesses include the somewhat anecdotal scale of the findings (GVzw, wkRc, 4KN7), and the fact that the mitigation is trivial (filtering and formatting the text prompt) (GVzw, k5Mu, wkRc). Reviewers dispute the terminology “attack” because it can be easily identified (k5Mu, wkRc, 4KN7). Reviewer wkRc had concerns about the evaluation itself.

Based on the scores (1, 3, 5, 6), this paper does not meet the publication bar. A comment-based discussion took place between three reviewers, including the one with the lowest score, and converged on some weaknesses of the claims even if they found merit to the paper. This paper could perhaps find a good publication venue in a conference or workshop focused on bias, ethics or creative applications of generative models.

**Summary Of Ac-Reviewer Meeting:**

N/A